# Spectroscopic and Petrographic Investigations of Lunar Mg-Suite Meteorite Northwest Africa 8687

Lang Qin [1,2], Xing Wu [1,*], Liying Huang [1,2], Yang Liu [1,3] and Yongliao Zou [1]

1   State Key Laboratory of Space Weather, National Space Science Center, Chinese Academy of Sciences, Beijing 100190, China; qinlang@nssc.ac.cn (L.Q.); huangliying21@mails.ucas.ac.cn (L.H.); yangliu@nssc.ac.cn (Y.L.); zouyongliao@nssc.ac.cn (Y.Z.)
2   University of Chinese Academy of Sciences, Beijing 100049, China
3   Center for Excellence in Comparative Planetology, Chinese Academy of Sciences, Hefei 230026, China
*   Correspondence: wuxing@nssc.ac.cn

**Abstract:** Magnesian suite (Mg-suite) rocks represent plutonic materials from the lunar crust, and their global distribution can provide critical information for the early magmatic differentiation and crustal asymmetries of the Moon. Visible and near-infrared (VNIR) spectrometers mounted on orbiters and rovers have been proven to be powerful approaches for planetary mineral mapping, which are instrumental in diagnosing Mg-suite rocks. However, due to the scarcity and diversity of Mg-suite samples, laboratory measurements with variable proportions of minerals are imperative for spectral characterization. In this study, spectroscopic investigation and petrographic study were conducted on lunar Mg-suite meteorite Northwest Africa 8687. We classify the sample as a pink spinel-bearing anorthositic norite through spectral and petrographic characteristics. The ground-truth information of the Mg-suite rock is provided for future exploration. Meanwhile, the results imply that the VNIR technique has the potential to identify highland rock types by mineral modal abundance and could further be applied in extraterrestrial samples for primary examination due to its advantage of being fast and non-destructive.

**Keywords:** meteorite; Moon; spectroscopy; Mg-suite; anorthositic norite



## 1. Introduction

Lunar meteorites and samples returned by the Apollo and Luna missions and the recent Chang'E-5 program provide significant insights into the formation and evolution of the Moon (e.g., [1–3]). Beyond that, large-scale remote-sensing observations have greatly revolutionized our understanding of the unsampled place (e.g., [4]). Early evolutionary models based on sample studies predicted the existence of a globally distributed layer enriched in incompatible elements (e.g., potassium (K), rare earth elements (REE), and phosphorus (P); referred to as KREEP) produced by the fractional crystallization of the lunar magma ocean (LMO) [5,6]. However, global remotely sensed data reveal that only a restrictive area on the lunar nearside shows KREEPy enrichment. This distinct province is the so-called Procellarum KREEP Terrane (PKT), since the region is coincidentally located in the Procellarum and Imbrium impact basin [4,7,8]. The lateral extent and distribution of KREEP are controversial, and the crustal asymmetry would be essential for understanding the planetary differentiation at large scales.

Ancient crustal samples bear critical information on lunar magmatic differentiation [3]. These nonmare igneous rocks, which record the early history of the Moon, mainly include the ferroan anorthosites (FANs) and magnesian suite (Mg-suite) rocks [9–11]. The FANs formed the primary crust by accumulating from the global magma ocean and are the main components (probably 80%) of lunar highlands [12]. Mg-suite rocks, also known as Mg-rich plutonic rocks, likely represent materials from the lower lunar crust [3]. Due to their deep crustal origin, few outcrops are present on the lunar surface, and the Mg-rich rocks account

for merely a tiny proportion of lunar samples (around 0.7% of 381.7 kg samples returned by the Apollo programs [10]). Despite this, the global distribution of Mg-suite rocks is crucial for constraining its petrogenesis and the last evolutionary stage proposed by the LMO hypothesis [13]. Early research on the Apollo returned sample suggested that Mg-suite magmatism was strongly controlled by KREEPy magma [3,13,14]. Namely, the distribution should be closely associated with the geochemical anomalous PKT [4]. Nevertheless, the trace elemental gap between lunar meteorites and the Apollo samples [15–17], along with new laboratory models [18], indicate that high magnesian rocks can be crystallized from KREEP-free melts; that is, the distribution of Mg-suite rocks could be Moon-wide. Whether Mg-suite rocks are widespread and their petrogenetic link to the last dregs of magma ocean holds the keys to unlocking the mysteries of their origin and asymmetries on the planetary body, while it remains an unsolved enigma in lunar science.

The possible Mg-suite materials on the lunar surface can be identified by discerning varied pyroxene- and olivine-bearing assemblages and Mg-spinel [13,19–21]. Visible and near-infrared (VNIR) spectroscopy has the potential to differentiate various minerals [13,22]; hence, it has attracted increasing attention over the past decades. Several spectral surveys identified Mg-suite clusters in the eastern South Pole–Aitken (SPA) and the northern rim of the Imbrium Basin [13]. Cahill et al. [23] detected Mg-suite exposures in central peaks of craters. However, the identical spectra could lead to different interpretations. For instance, the mineral compositions of lunar rocks in the SPA decoded from the VNIR spectra of the Yutu-2 rover yielded significant differences [24,25]. It is challenging to validate the mapping results. The ground-truth information is essential in calibrating and interpreting spectral features, while extensive laboratory measurements of variable proportions of minerals are lacking, due to the scarcity and diversity of Mg-suite rocks. Furthermore, previous laboratory spectral studies have mainly focused on bulk samples in powder form, which is difficult to interpret the spectral characteristics of individual clasts precisely [26].

In contrast to returned samples from limited areas, lunar meteorites represent random samplings of the lunar surface [1]. This allows meteoritic samples to contribute to a more diverse lithological dataset. Moreover, due to the rarity of extraterrestrial samples, non-destructive methods need to be explored for future analysis. To achieve these aims, we followed the efforts of [26–28], and conducted a spectral survey on lunar meteorite Northwest Africa (NWA) 8687. To well-characterize the sample, a careful petrographic study was performed. Some of the highlights of this study are as follows:

- A new compilation of mineral chemistry of NWA 8687 is reported. Based on detailed analyses of mineral composition and spectral investigation, the sample NWA 8687 is confirmed as a lunar Mg-suite anorthositic norite.
- The coordinated reflectance spectra of NWA 8687 are acquired. Unlike the VNIR spectra of bulk samples collected in powder form, the measurement of NWA 8687 was performed on a chip, which facilitates spectral analysis of individual rocks and clasts.
- The VNIR spectroscopy is helpful for both the in situ characterization of lunar rocks and non-destructive laboratory characterization of returned samples and lunar meteorites.

## 2. Materials and Methods

### 2.1. Lunar Meteorite Northwest Africa 8687

Northwest Africa (NWA) 8687 was found in a hot desert within Morocco in April 2014, and it was broken into 5 pieces, weighing a total of 563 g [29]. Based on textural and chemical similarities, NWA 8687 is likely to be paired to the lunar granulitic anorthositic troctolite breccia NWA 5744 clan (including paired stones NWA 5744, NWA 8599, NWA 10140, NWA 10178, NWA 10318, NWA 10401, and NWA 11252) [30–33].

The saw-cut surface of the sample used for point spectroscopy analysis in this study is greenish-white, with large mineral grains and dark impact melt veins visible (Figure 1). After obtaining the spectral data, a 1-inch polished thin section made with the same chip was prepared for petrological analysis, and the area for research is around 16 × 14 mm.

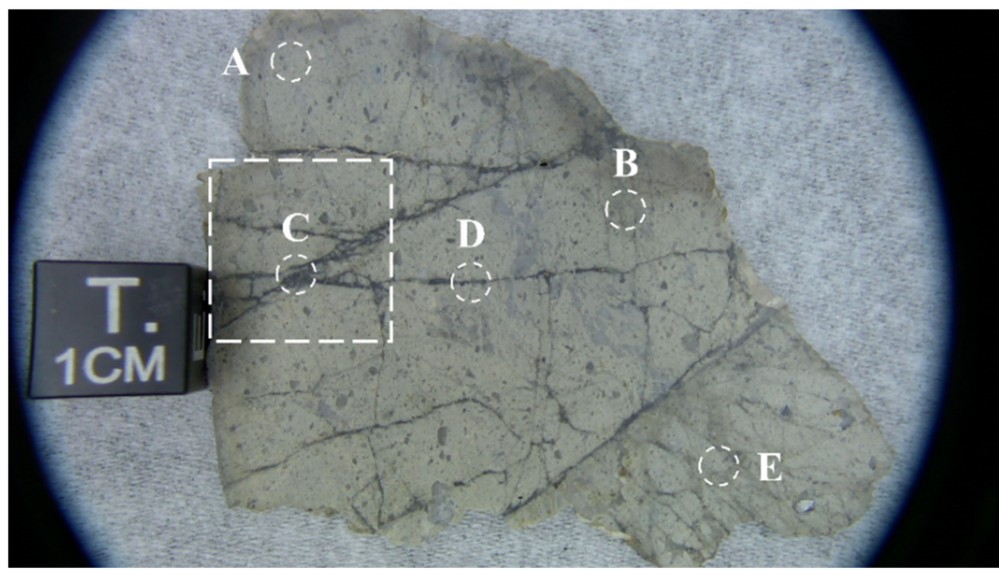

**Figure 1.** Photograph of lunar meteorite Northwest Africa 8687 and the location of point spectral measurements (**A**–**E**) in this study. The frame shows the rough location of the polished thin section for the petrographic study.

### 2.2. Visible and Near-Infrared Reflectance Spectroscopy

The VNIR spectra were acquired by the PSR-3500 portable spectrometer (Spectral Evolution Inc., Lawrence, MA, USA), which covers a spectral range of 350–2500 nm, and offers a spectral resolution of 3.5 nm at 700 nm, 10 nm at 1500 nm, and 7 nm at 2100 nm [34]. The number of spectral bands is 1024. The spectralon produced by Labsphere was used to provide the absolute reflectance factor for spectral measurements. A 50 W halogen lamp mounted 60 cm above the samples served as the light source in the darkroom. To facilitate comparison with other lunar samples' spectra, i.e., spectra from the Reflectance Experiment Laboratory (RELAB) database, we set the incidence and emergence angles as 30° and 0°, respectively. A total of 10 scans were averaged for each spectrum in order to improve the signal-to-noise ratio. The fields of view of five spectral measurements are shown in Figure 1.

### 2.3. Petrography and Mineralogical Analysis

2.3.1. Scanning Electron Microscope

The backscattered electron (BSE) image of the thin section of NWA 8687 was captured by the Thermofisher Apreo field emission scanning electron microscope (FE-SEM) at the Institute of Geology and Geophysics, Chinese Academy of Sciences (IGGCAS). The instrument is equipped with a QUANTAX energy dispersive spectrometer (EDS) for obtaining Kα X-ray element maps under 15 kV and 13 nA operating conditions. The X-ray acquisition time is 8 ms per pixel, and the spacing is 10 μm.

Mineral modal abundance is achieved by automated mineral identification based on X-ray maps via the software NanoMin. To compare the result, we used the software ImageJ to count the gray values of the BSE image, based on the principle that the values are related to chemical compositions, and can be used as proxies for minerals.

2.3.2. Electron Probe Micro Analyzer

The quantitative analysis of main mineral phases was carried out by electron probe microanalysis (EPMA) using the JEOL JXA-8100 at the Institute of Geology, Chinese Academy of Geological Sciences (CAGS). The acceleration voltage of the EPMA is 15 kV, and the beam current is 20 nA. The focused beams measured 1 to 5 μm, adjusted by the size of the mineral grains. Natural minerals of known compositions were used as standards for the

corresponding minerals, and the standard used for impact melt veins was consistent with plagioclase. All data were corrected using the ZAF method.

The bulk composition was calculated by the weight percentage of mineral phases and their average point analysis, using the equation:

$$Bulk = \frac{\sum \chi \rho c}{\sum \chi \rho} \tag{1}$$

where $\chi$ = mineral phase abundance; $\rho$ = mineral phase density; $c$ = average element concentration.

*2.4. Spectral Analysis*

2.4.1. Spectral Parameters

The acquired spectrum was first smoothed using the Savitzky–Golay filter with a window size of 11 bands and a polynomial order of 2 [35]. The continuum removal was conducted to highlight absorption features while removing the baseline [36]. The band center and area are widely used to differentiate the mafic minerals [37,38]. To estimate the exact band center and area, we employed a sixth-order polynomial fitting to the absorption regions around ~1 μm and ~2 μm of the continuum-removed spectra [39]. The band center is defined as the wavelength with the minimum value according to the polynomial function. The band area is defined as the area covered by the absorption band of a given absorption feature. The band area ratio (BAR) is the ratio of the Band II (2 μm) area to the Band I (1 μm) area.

2.4.2. Spectral Similarity Measurement

We collected all available RELAB reflectance spectra of lunar samples and searched for the best matches between NWA 8687 spectra and the library spectra. The spectral angle mapper (SAM) is a tool that permits the rapid mapping of spectral similarity. The SAM algorithm (Equation (2)) determines the similarity of two spectra by calculating the angle parameter θ (in degree) between them [40]. Smaller spectral angles indicate a higher similarity between the NWA 8687 spectra and the laboratory spectra. The reason for choosing the SAM technique is that it concentrates on the spectral shape and is insensitive to absolute reflectance associated with illumination or albedo effects.

$$\theta = \cos^{-1}\left(\frac{\sum_{i=1}^{L} x_i y_i}{\sqrt{\sum_{i=1}^{L} x_i^2}\sqrt{\sum_{i=1}^{L} y_i^2}}\right) \tag{2}$$

where $x$ and $y$ are spectra from the RELAB database and NWA 8687, respectively.

As the spectra of NWA 8687 and the RELAB database were measured using different spectrometers, we performed the spectral resampling first to ensure the success of the comparison using SAM. A total of 884 RELAB spectra of returned samples and lunar meteorites have been investigated. We ranked the matching results in ascending order of the spectral angle. The spectrum with the smallest spectral angle was selected as the best match.

2.4.3. Spectral Unmixing

The signal collected by a spectrometer is often a mixture of light scattered by substances located in the field of view [41]. To separate the observed spectrum into a collection of pure endmember spectra and their respective abundances, both the linear and nonlinear mixing models were proposed. The reflectance spectra of meteorite chip surfaces can be regarded as linear mixtures, since minerals are separated from one another on a much larger scale

than the average grain size and wavelength, so that reflected light from different minerals do not interact significantly [27]. The observed spectrum $y$ can be represented as follows:

$$y = M\alpha + n \tag{3}$$

where $y \in R^{L \times 1}$ is the spectral vector with $L$ bands, $M \in R^{L \times q}$ is the endmember matrix containing $q$ pure endmembers, $\alpha \in R^{q \times 1}$ is the abundance vector, and $n \in R^{L \times 1}$ denotes the additive perturbation due to spectral noise and modeling error. The abundance vector should follow two physical constraints, namely the abundance nonnegativity constraint (ANC) and the abundance sum-to-one constraint (ASC):

$$\alpha \geq 0 \tag{4}$$

$$\mathbf{1}^T \alpha = 1 \tag{5}$$

where $\mathbf{1}^T$ is a line vector of 1's compatible with $\alpha$.

In this work, we used the linear unmixing model with two physical constraints [42] to derive mineral abundances. Plagioclase, pyroxene, olivine, and ilmenite were used as endmembers, for the reason that they are the four major minerals of the Moon [43]. The pyroxene contains clinopyroxene and orthopyroxene, and the plagioclase includes plagioclase glass (maskelynite). The endmembers were manually selected based on the petrographic results and are exhibited in Figure 2. The spectra of endmembers were collected in powder form, while that of NWA 8687 were in chip form. For meteorite chips, the reflectance of the left shoulder of 1 μm absorption is greater than the right shoulder of 2 μm absorption. Thus, measuring the samples as chips rather than particulates led to a pronounced negative continuum slope [26]. The continuum removal was performed on both the endmember and NWA 8687 spectra first, and then the linear unmixing was conducted on the continuum removed spectra.

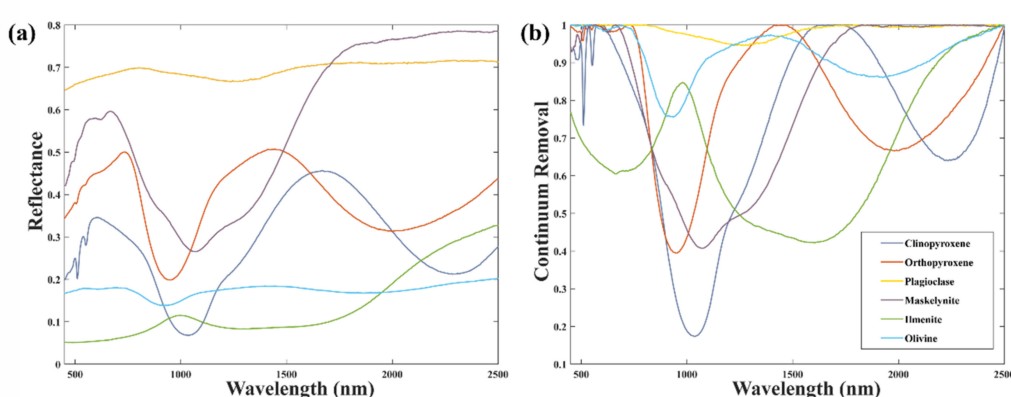

**Figure 2.** (**a**) Visible and near-infrared (VNIR) reflectance spectra and (**b**) spectra after continuum removal of endmembers used for spectral unmixing. Spectral ID: clinopyroxene (C1DL84A), orthopyroxene (C1LR209), plagioclase (C1LR223), maskelynite (C1LR222), ilmenite (C1LR222), and olivine (C2PO81).

## 3. Results

### 3.1. Petrography and Mineral Modal Abundance

The meteorite NWA 8687 shows a porphyritic texture, with large phenocrysts embedded in the fine-grained matrix and impact melt veins crosscutting the samples (Figure 3). Few clasts are present in the sample, indicating that NWA 8687 is a monomict breccia.

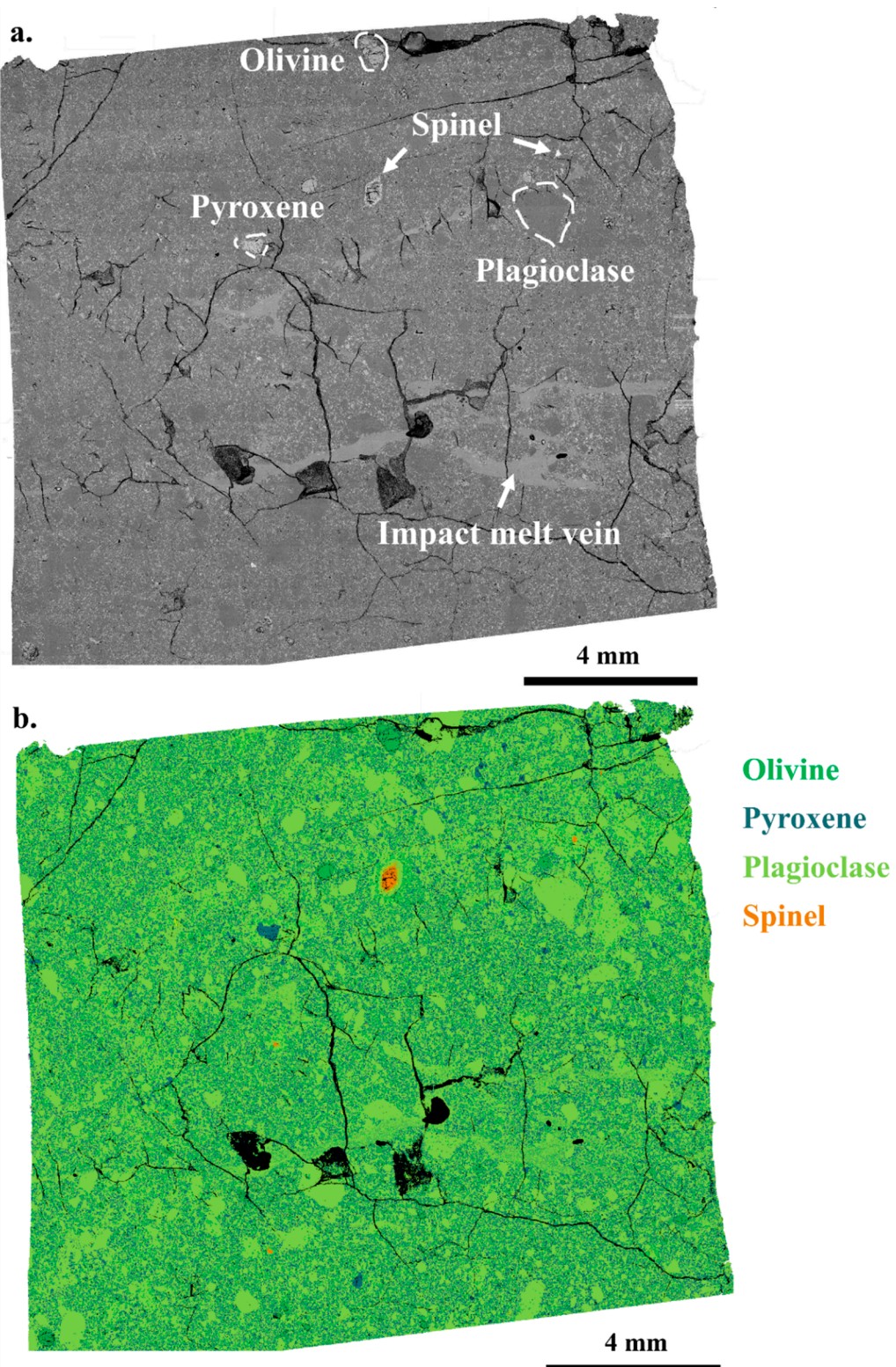

**Figure 3.** (**a**) backscattered electron (BSE) image of a polished section of NWA 8687. (**b**) Distribution of main minerals in NWA 8687 based on X-ray element maps.

The sample in this study mainly consists of plagioclase (or maskelynite; 67.1–71.8 vol.%), pyroxene (16.5–20.8 vol.%), and olivine (11.6–11.8 vol.%), along with minor spinel (0.1–0.2 vol.%) (Table 1). Accessory minerals include Fe–Ni metals (0.07 vol.%), troilites

(0.04 vol.%), ilmenites (0.04 vol.%), and baddeleyites (0.0002 vol.%). Compared with pairs NWA 5744 and NWA 10401 [15,32], NWA 8687 has more pyroxene and less olivine.

**Table 1.** Mineral modal abundance in the lunar meteorites NWA 8687 compared to pairs NWA 5744 and NWA 10401 (in vol. %).

| Method [3] | NWA 8687 | | NWA 5744 [1] | NWA 10401 [2] | |
| --- | --- | --- | --- | --- | --- |
| | NanoMin | BSE | BSE | XMapTool | IDRISI Selva |
| Plagioclase * | 67.10 | 71.8 | 67 | 59 | 65 |
| Olivine | 11.80 | 11.6 | 26.3 | 26 | 23 |
| Total pyroxene * | 20.77 | 16.5 | 6.6 | 15 | 12 |
| High-Ca pyroxene | 1.03 | - | - | - | - |
| Low-Ca pyroxene | 19.74 | - | - | - | - |
| Spinel/chromite | 0.18 | 0.1 | 0.1 | 0.001 | 0.001 |
| Metal | 0.07 | - | - | - | - |
| Troilite | 0.04 | - | - | - | - |
| Ilmenite | 0.04 | - | - | - | - |
| Baddeleyite | 0.0002 | - | - | - | - |
| Total | 100 | 100 | 100 | 100 | 100 |

* Plagioclase includes plagioclase glass (maskelynite). Total pyroxene is the sum of high-Ca pyroxene and low-Ca pyroxene. [1] Data from Ref. [32]. [2] Data from Ref. [15]. [3] Abbreviations represent modal abundance calculated by NanoMin, XMapTool, IDRISI software, or by backscattered electron (BSE) image grayscale statistics.

### 3.2. Mineral Chemistry

Major elements of the four major mineral phases in NWA 8687 were analyzed in this study. The representative analyses of individual minerals and estimated bulk composition are given in Table 2 and Supplementary Tables S1–S5.

**Table 2.** Representative EPMA analysis of olivine, pyroxene, plagioclase (or maskelynite), spinel, and impact melt vein, and estimated bulk composition of lunar meteorite NWA 8687 (in wt.%).

| $n$ | Olivine | | | Pyroxene | | Plagioclase | | Spinel | | Vein | Bulk [1] |
| --- | --- | --- | --- | --- | --- | --- | --- | --- | --- | --- | --- |
| | Grain | Matrix | Vein | Low-Ca | High-Ca | Grain | Matrix | Mg-Al | Cr-Fe | | |
| | 17 | 9 | 5 | 14 | 12 | 8 | 6 | 5 | 4 | 6 | |
| $SiO_2$ | 38.73 | 38.69 | 38.65 | 53.96 | 51.31 | 45.03 | 44.78 | 0.46 | 0.45 | 46.03 | 46.16 |
| $TiO_2$ | 0.04 | 0.08 | 0.06 | 0.69 | 1.30 | 0.02 | 0.03 | 0.46 | 5.41 | 0.29 | 0.20 |
| $Al_2O_3$ | 0.08 | 0.12 | 0.02 | 1.34 | 2.09 | 34.84 | 35.04 | 57.36 | 19.53 | 23.41 | 22.36 |
| $Cr_2O_3$ | 0.04 | 0.04 | 0.04 | 0.40 | 0.67 | 0.01 | 0.04 | 8.93 | 37.78 | 0.15 | 0.17 |
| $V_2O_3$ | - | - | - | - | - | - | - | 0.23 | 0.51 | - | 0.001 |
| FeO | 20.98 | 21.15 | 20.60 | 12.52 | 7.77 | 0.13 | 0.18 | 15.59 | 26.85 | 5.40 | 5.82 |
| MnO | 0.23 | 0.23 | 0.23 | 0.23 | 0.18 | 0.01 | 0.01 | 0.13 | 0.30 | 0.08 | 0.09 |
| MgO | 39.26 | 38.94 | 39.83 | 27.47 | 16.31 | 0.12 | 0.08 | 16.42 | 8.44 | 10.28 | 11.66 |
| CaO | 0.12 | 0.17 | 0.08 | 2.64 | 19.81 | 19.17 | 19.03 | 0.07 | 0.12 | 13.65 | 12.85 |
| NiO | 0.01 | 0.01 | 0.01 | - | - | - | - | - | - | - | 0.001 |
| CoO | 0.04 | 0.04 | 0.04 | - | - | - | - | - | - | - | 0.01 |
| $Na_2O$ | 0.01 | 0.02 | 0.02 | 0.02 | 0.08 | 0.36 | 0.37 | - | - | 0.27 | 0.24 |
| $K_2O$ | 0.01 | 0.01 | 0.01 | 0.01 | 0.01 | 0.01 | 0.02 | - | - | 0.01 | 0.01 |
| $P_2O_5$ | 0.01 | 0.02 | 0.01 | 0.02 | 0.02 | 0.01 | 0.02 | - | - | 0.03 | 0.02 |
| Total | 99.56 | 99.53 | 99.62 | 99.31 | 99.55 | 99.72 | 99.61 | 99.66 | 99.39 | 99.59 | 99.57 |

[1] Bulk composition is estimated by Equation (1).

### 3.2.1. Pyroxene

Pyroxene is the most abundant mafic mineral in NWA 8687. It appears throughout the sample as orthopyroxene (OPX; 19.74 vol.%) and a small amount of clinopyroxene (CPX; 1.03 vol.%). These two phases belong to low-Ca pyroxenes and high-Ca pyroxene, respectively. The CPX presented as phenocrysts in the slab, along with OPX, whereas no exsolution texture was seen. In the matrix, low-Ca pyroxene and olivine usually grew together, surrounded by anhedral plagioclase glass (maskelynite) (Figure 4c).

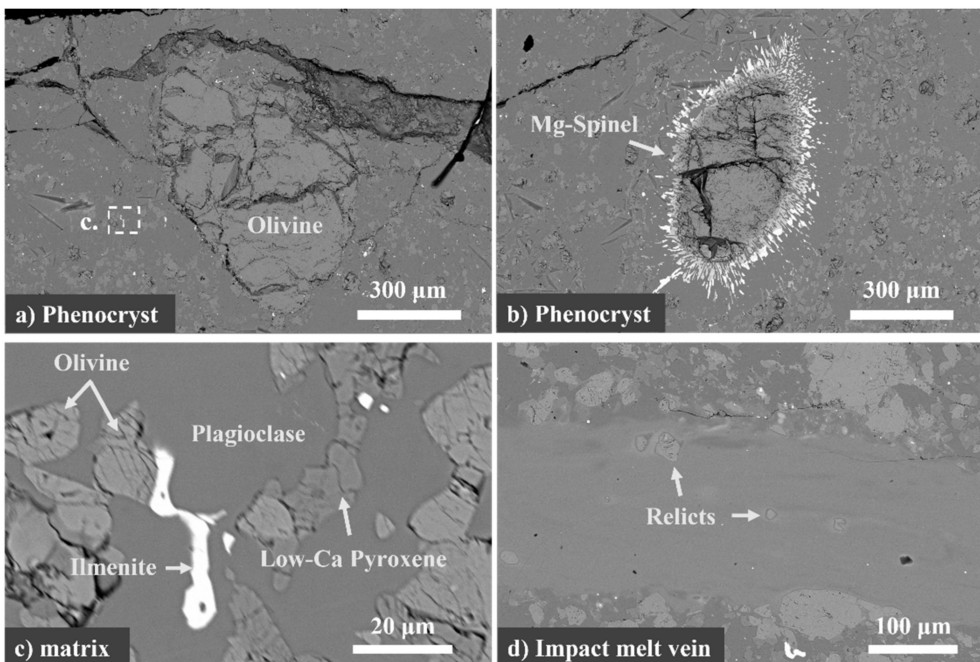

**Figure 4.** Backscattered electron (BSE) images of (**a**) an olivine phenocryst. The frame shows the location of figure (**c**). (**b**) A Mg-spinel in the matrix. (**c**) Fine-grained matrix, and (**d**) some mineral relics are visible in the impact melt vein.

The composition of OPX in NWA 8687 is Mg-rich ($En_{65-78}$ $Fs_{17-20}$ $Wo_{3-20}$; $n$ = 14), while the CPX has a compositional range of $En_{43-50}$ $Fs_{10-16}$ $Wo_{38-43}$ ($n$ = 12) (Figure 5a). The Mg number (Mg#; molar Mg/(Mg + Fe) × 100) in pyroxene ranges from 72.8 to 83.0, with an average of 79.3. The pyroxene composition of NWA 8687 is consistent with that of NWA 5744 and NWA 10401 [15,32], apart from a slightly lower calcium content.

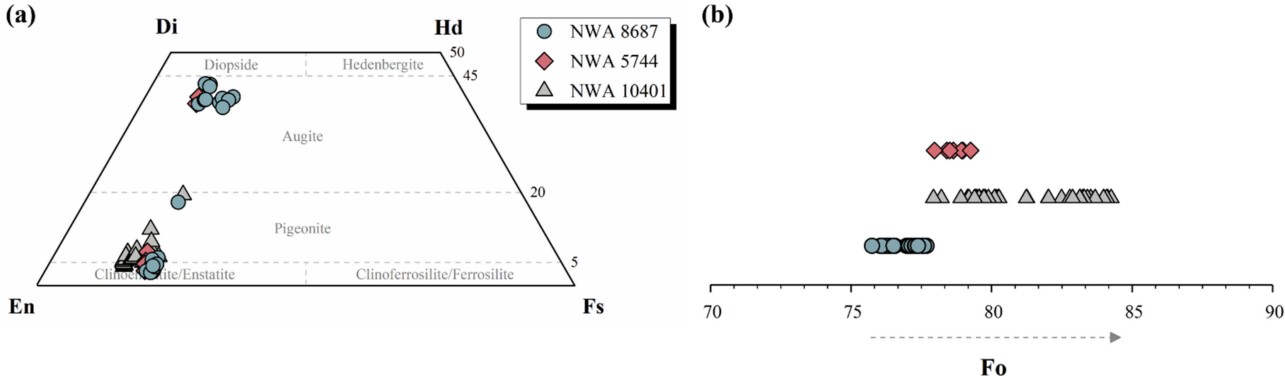

**Figure 5.** Pyroxene (**a**) and olivine (**b**) composition for NWA 8687, compared with a counterpart from pairs NWA 5744 and NWA 10401 [15,32]. Di—diopside, Hd—hedenbergite, En—enstatite, Fs—ferrosilite, and Fo—forsterite.

### 3.2.2. Olivine

Phenocryst olivine was distributed occasionally in the matrix, with grain sizes up to 700 μm (Figure 4a). Fine-grained olivine was also ubiquitous in the matrix, with varying sizes of several to tens of micrometers (Figure 4c). Cracks were on the surface of each olivine grain.

The composition of olivine in NWA 8687 is identical and forsteritic, with a high Mg# of 76.9 on average. The forsterite (Fo) contents of the 31 selected olivine grains varied

between 75.8 to 77.7 mol%, with an average of 76.9 ± 0.6 mole%. However, compared to the NWA 5744 clan, olivine in NWA 8687 is relatively depleted in Mg (Figure 5b).

### 3.2.3. Plagioclase and Maskelynite

The feldspar in NWA 8687 is predominantly plagioclase (Ca-rich feldspar), with a narrow range of compositions ($An_{95.6-97.5}$ $Ab_{2.4-4.2}$ $Or_{0-0.2}$) (*n* = 14). Some of them exist in the sample as phenocryst (plagioclase) (Figure 3), and some are devitrified to glass (maskelynite). This article will not distinguish these two phases particularly.

### 3.2.4. Spinel

Spinel is a general term for a series of oxide minerals with diverse compositions, which can be represented by the spinel multicomponent prism [44]. It is common in mare basalts, and most lunar spinel is iron-rich, with components falling between chromite and ilmenite [43]. Remote-sensing observations have shown that Mg-rich spinel (also known as pink spinel) are ubiquitous on the lunar surface [20,21], while spinel of this composition is rare in lunar samples, most of which are highland rocks (e.g., pink spinel troctolite) [43].

A magnesian spinel surrounded by chromites and ilmenites in NWA 8687 is shown in Figure 4b. The Cr# (molar Cr/(Cr +Al) × 100) and Mg# (molar Mg/(Mg + Fe) × 100) of the Mg-spinel are in the range of pink spinel (Figure 6), although it still does not reach the component observed by the $M^3$ (Mg# > 90, Cr# < 5) [21].

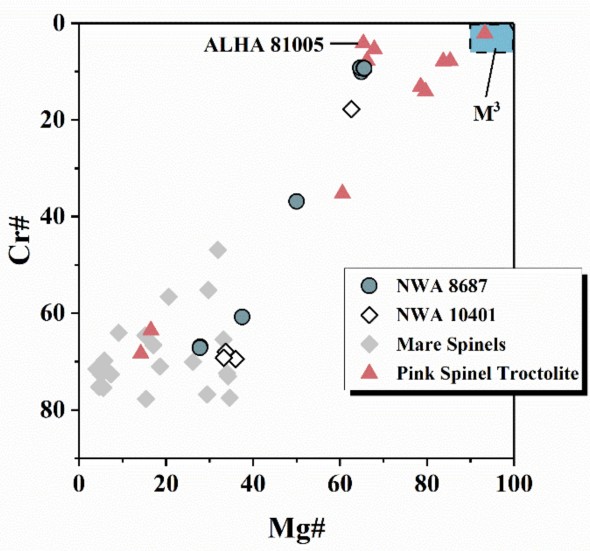

**Figure 6.** Spinel composition in NWA 8687 relative to compositional range from $M^3$ (after Ref. [21]); also compared with spinel in pair stone NWA 10401 [15], as well as in mare basalts [45] and pink spinel troctolites [46–54].

### 3.2.5. Impact Melt Veins

Several impact melt veins presented as crosscutting the NWA 8687, and there were mineral relics in veins that were not completely melted (Figure 4d). A total of six spots were analyzed in the study, and the result was shown to be chemically homogenous. Notably, the average composition of these impact melt veins is almost consistent with that of the entire rock (Table 2), with an Mg number of 78.

### 3.3. Minerlogy Inferred from VNIR Spectroscopy

### 3.3.1. Qualitative Analysis

As demonstrated in Figure 7, the VNIR spectra of NWA 8687 are characterized by two broad and strong absorptions at ~900 nm and ~1900 nm, demonstrating the presence of low-Ca pyroxene. The weak absorption at ~1250 nm indicates the occurrence of abundant

plagioclase in NWA 8687. This feature can be easily muted by mafic minerals. For example, the spectral mixture model shows that only 2 vol.% pyroxene could mask the ~1250 nm absorption in plagioclase [55]. The weak absorption at ~1300 is indicative of olivine.

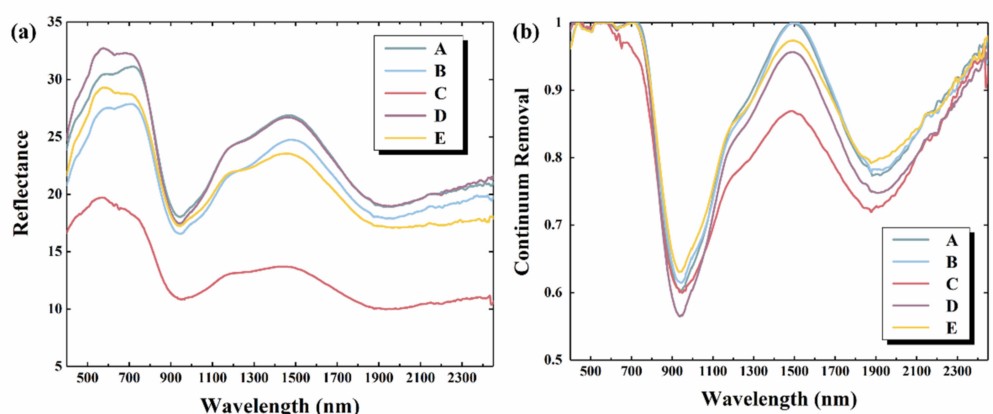

**Figure 7.** (**a**) Visible and near-infrared (VNIR) reflectance spectra of NWA 8687 chip surface spots marked in Figure 1. (**b**) VNIR spectra after continuum removal.

The best-matched spectra are S3LS07 and CFLM34 (spectrum ID), with the spectral angle of 1.91° and 3.18°, taken separately from the returned sample 60019.214, and lunar meteorite Allan Hills A81005 (ALHA 81005) (Figure 8a). Sample 60019 was returned by the Apollo 16 mission, and it has been classified as an ancient regolith breccia [56]. According to the geochemistry data documented by RELAB, the Mg number of sample 60019.214 is 69.2, together with an $Al_2O_3$ content of 26.3%, indicating that the sample has the characteristics of FANs instead of Mg-suite rocks. The ALHA 81005 is a regolith breccia with various clasts, such as basalts, FANs, and Mg-suite rocks [57], whereas the mineral modal abundance and composition of the specified sample used for collected spectra are not available.

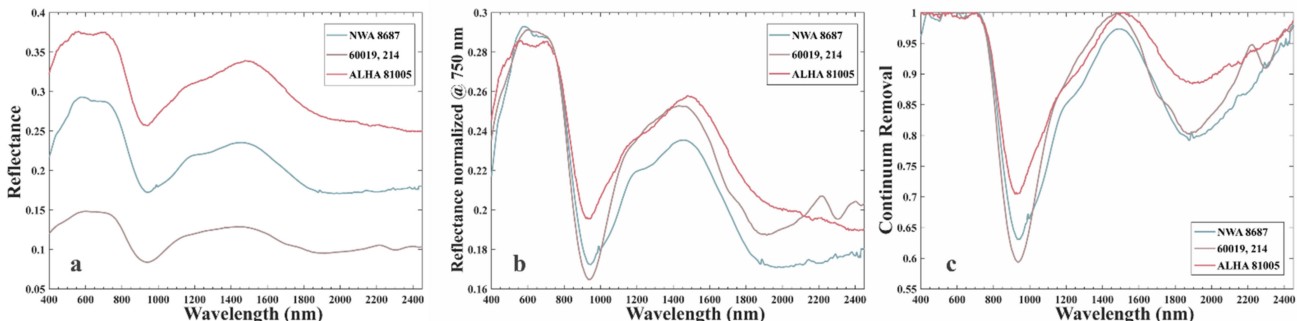

**Figure 8.** Comparisons of NWA 8687 and the best-matched RELAB reflectance spectra of returned lunar sample 60019, 214, and lunar meteorite ALHA 81005 using SAM algorithm. (**a**) Reflectance. (**b**) Spectra normalized at 750 nm, and (**c**) spectra after continuum removal. Spectrum ID for sample 60019.214 and ALHA 81005 is S3LS07 and CFLM34, respectively.

For visual comparison, the NWA 8687 spectrum and the RELAB spectra were normalized at 750 nm (Figure 8b). The continuum removal was performed (Figure 8c) to compare the band centers and depths. Since the reflectance of the left shoulder of 1 μm absorption is greater than the right shoulder at 2 μm absorption, the continuum shows a negative slope. Similar absorption features near ~900 nm and ~1900 nm were demonstrated in three reflectance spectra, while NWA 8687 has deeper absorptions. Additionally, ALHA 81005 and NWA 8687 show absorption at ~1250 nm, indicating that they are rich in plagioclase.

Figure 9a illustrates the Band I and Band II centers of NWA 8687 spectra in the background of those produced from laboratory synthetic pyroxene [58]. Generally, adding

Ca into pyroxene crystal makes Band I and Band II centers shift to longer wavelengths [38]. The OPX cluster is at the lower left corner, while the CPX is mainly distributed in the upper right corner. Moreover, the high Mg (Mg# > 70) pyroxene is annotated in the shaded region, based on laboratory results. Band centers of NWA 8687 are in the range of OPX, and are located in the high Mg region, indicating that the composition of pyroxene is low in calcium and high in magnesium, which is in agreement with the petrography.

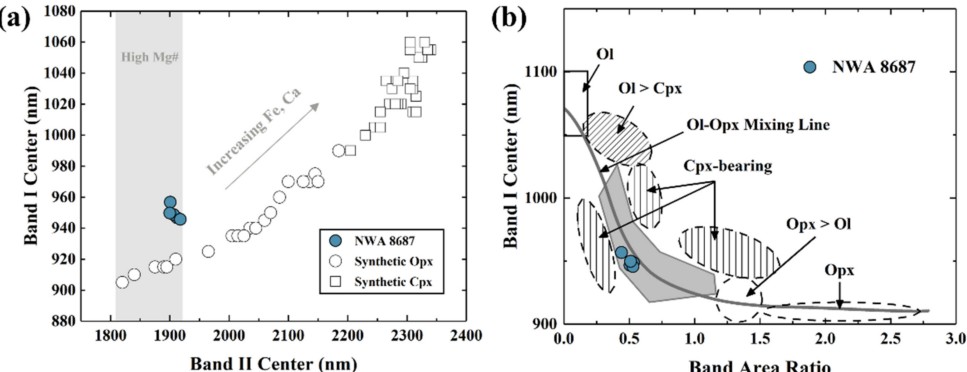

**Figure 9.** (**a**) Plot of the Band I center versus the Band II center. The laboratory pyroxene spectra are from Ref. [58]. (**b**) Plot of the Band Area Ratio versus the Band I center. The olivine–orthopyroxene mixing line is represented by the solid line. Different spectral regions are defined by Refs. [37,59].

The band area ratio is inversely proportional to olivine abundance [59], and its relationship to Band I center is useful for estimating the relative abundance of olivine and OPX [37,59]. The NWA 8687 spectra allocate on the Ol–OPX mixing line (Figure 9b), suggesting that NWA 8687 contains both OPX and olivine, while the proportions of the two-phase are not well distinguished.

### 3.3.2. Quantitative Analysis

From the results of spectral unmixing (Table 3), the NWA 8687 is dominated by OPX and plagioclase (including meskelynite), with an average abundance of 54.02% and 34.05%, respectively. Olivine follows as the third most abundant constituent (7.08% on average). There are tiny amounts of CPX and ilmenite in the meteorite, consistent with the petrography (Table 1). Since OPX easily masks the spectrum of plagioclase, the derived OPX abundance is higher than that of plagioclase.

**Table 3.** Mineral abundances are estimated from NWA 8687 spectra (in %).

|  | A | B | C | D | E |
|---|---|---|---|---|---|
| Clinopyroxene | 0 | 1.51 | 0 | 0 | 1.71 |
| Orthopyroxene | 58.20 | 50.33 | 55.30 | 63.55 | 42.73 |
| Plagioclase | 35.24 | 28.49 | 20.83 | 24.82 | 24.53 |
| Olivine | 6.08 | 6.12 | 9.58 | 7.55 | 6.08 |
| Ilmenite | 0.49 | 0.50 | 14.28 | 4.07 | 1.69 |
| Maskelynite | 0 | 13.05 | 0 | 0 | 23.27 |
| Total | 100 | 100 | 100 | 100 | 100 |

Once the abundance vector is derived, the modeled spectrum can be obtained through the endmember matrix multiple by the abundance vector. The measured and modeled spectra generally fit well, with the exception of the shift around 1900 nm (Figure 10). The shift could be the difference in chemical compositions (mainly in Ca, Fe, and Mg) between the pyroxene in NWA 8687 and the pyroxene endmembers used in the unmixing process.

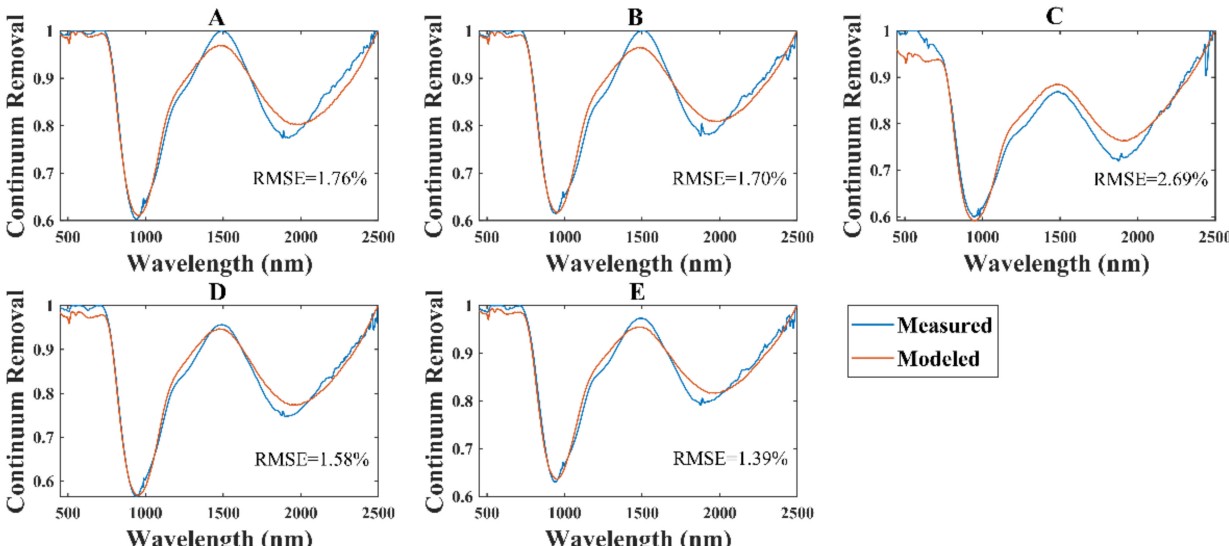

**Figure 10.** Comparisons between the measured point spectra (**A**–**E**) (in blue) and their modeled spectra (in red) of the NWA 8687 chip.

## 4. Discussion

### 4.1. Lunar Origin

Basalt magmatism is a basic geological process of the Moon, Earth, and other terrestrial planets. As probes to planetary interiors, basaltic silicate minerals record distinct igneous histories and signatures of their planetary bodies [60,61]. Figure 11 illustrates that each planet has a unique $Mn/Fe^{2+}$ ratio in pyroxene and olivine, and the ratio increases with the heliocentric distance, probably due to heating events in the early solar system [62,63]. The practically constant values are widely used as a "fingerprint" of meteorite origins (e.g., [15,64]). NWA 8687 has an average $Mn/Fe^{2+}$ ratio of 0.021 in pyroxene and 0.011 in olivine; both fall into the range of lunar samples (Figure 11). Therefore, we confirm that the sample in this study has a lunar origin.

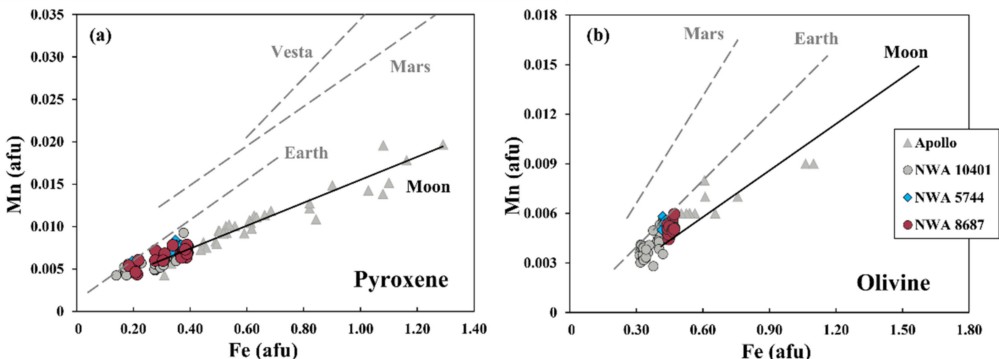

**Figure 11.** Mn versus $Fe^{2+}$ atoms per formula unit (afu) in pyroxene (**a**) and olivine (**b**) of NWA 8687, compared with pairs NWA 5744 [32] and NWA 10401 [15]. Apollo data and best-fit trend lines are from Ref. [60] (pyroxene) and Ref. [61] (olivine).

### 4.2. Classification of NWA 8687: A Mg-Suite Anorthositic Norite?

For mineral compositions, NWA 8687 is dominated by calcic plagioclase, with a high content of anorthite (An$_{95.6-97.5}$) coexisting with Mg-rich mafic silicates (Mg# between 75.8 and 77.7) (Figure 12a). This mineral assemblage is consistent with the Mg-suite rocks (e.g., [65,66]). For bulk chemistry, Mg number versus Al$_2$O$_3$ content of NWA 8687 and other lunar samples are plotted in Figure 12b (after Refs. [15,16,32]). The figure clearly shows that Mg-rich rocks are distinct from other lunar samples due to higher Mg# and lower Al$_2$O$_3$ (e.g., [67]), and NWA 8687 is located ideally in the Mg-suite region. Thus, NWA 8687 can be considered to have the characteristics of magnesian rocks.

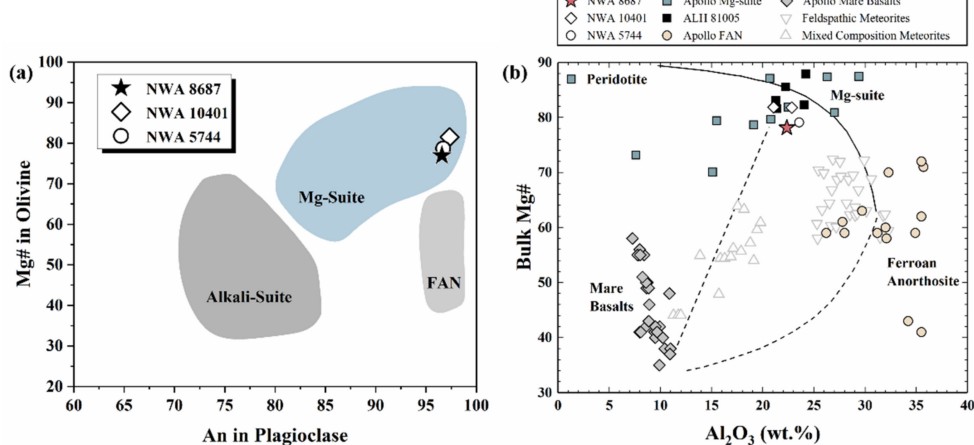

**Figure 12.** (**a**) Mg# in olivine versus An in plagioclase for NWA 8687 and pairs NWA 5744 and NWA 10401 (after Ref. [13]). (**b**) Bulk Mg# versus Al$_2$O$_3$ content for NWA 8687 and other lunar samples (after Refs. [15,16,32] and references therein). Lines show mixtures among four endmembers: a hypothetical magnesian peridotite, Mg-suite rocks, ferroan anorthosite, and mare basalts. Data source: NWA 5744 [32], NWA 10401 [15], ALH 81005 [16], and other lunar meteorites [33]; Apollo Mg-suite [68], FAN, and mare basalts [9].

For the convenience of academic communication, Stöffler et al. [69] developed a classification scheme to allocate lunar highland rocks, based on their modal compositions (Figure 13). The Mg-suite rock comprises norites, troctolites, ultramafic rocks, and closely related lithologies (e.g., anorthositic norite) in this scheme [15]. According to mineral modal abundances (Table 1; here, maskelynite is considered part of plagioclase), this particular sample, NWA 8687, is better described as an anorthositic norite instead of a troctolite, reported by previous studies [29,30,70]. Similarly, NWA 5744, examined by Ref. [31], also has noritic lithology differing from other studies. Although the meteorite and pairs NWA 5744 group have slight differences in classification, they are broadly the same in texture (Figures 3 and 4) and mineral chemistry (Figures 5, 6, and 12a), as well as whole-rock composition (Figure 12b); thus, NWA 8687 is still intimately connected with the NWA 5744 clan.

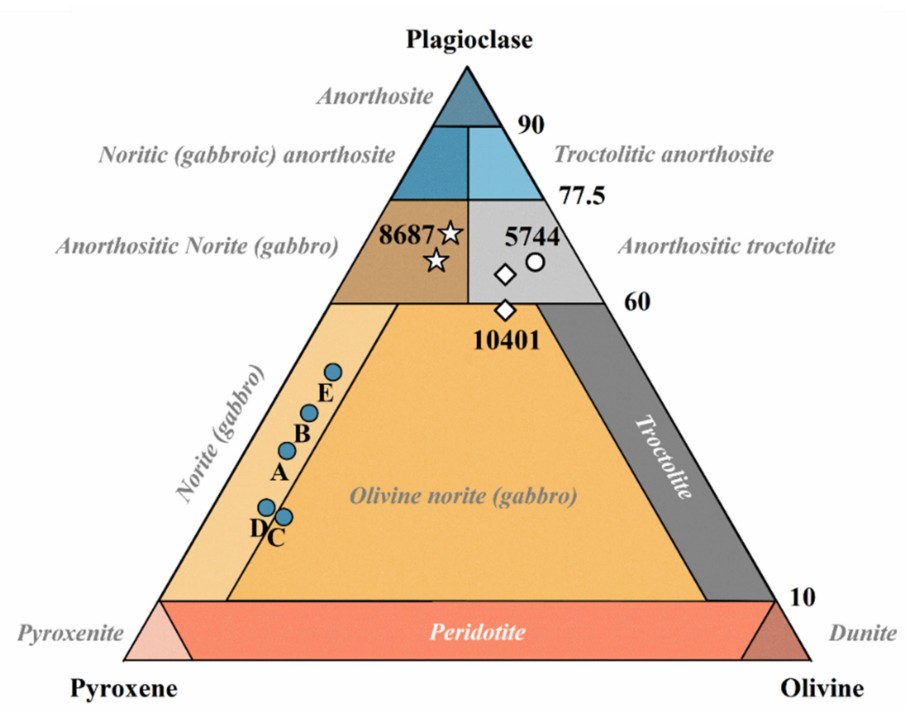

**Figure 13.** Classification of lunar meteorite NWA 8687 and comparison with pairs NWA 5744 and NWA 10401. Modal abundance (in vol.%) obtained by petrography (star) and by spectroscopy (blue circle) is expressed in the plot. This figure is modified from Figure 2 in Ref. [69].

*4.3. Connections between Lab Results and Remote-Sensing Interpretations*

The mineral abundances derived from spectral unmixing (Table 3) were compared with the petrographic ground-truth (Table 1). The average differences between the derived abundances and the ground-truth are −0.004 (CPX), 0.34 (OPX), −0.33 (plagioclase), and −0.05 (olivine), respectively. The result shows that the VNIR spectra can accurately determine the abundances of CPX and Olivine. As the OPX overwhelms the plagioclase spectral features, the meteorite spectral features are dominated by the OPX. Specifically, the VNIR spectra overestimate the OPX abundance (more than 2.5 times the actual fraction of OPX) and underestimate the plagioclase abundance (about half the actual fraction of plagioclase). This phenomenon could also happen during lunar orbital observations.

Nonetheless, as shown in the pyroxene–olivine–plagioclase ternary diagram (Figure 13), both the petrographic and spectral results are located at the left edge of the triangle, indicating that the NWA 8687 is noritic instead of troctolitic. In lunar geology, norite and almost all of the anorthositic norite represent material from the lower lunar crust [71]. Thus, the deviation in classification based on petrography and spectroscopy does not lead to considerable differences in regional geologic interpretations.

Our work shows that the VNIR spectroscopy has the potential to crudely classify lunar rock types without sample preparation. The orbital hyperspectral image (HSI) can be used to search for Mg-Suite exposures across the Moon. However, the spectral interpretation of the lunar surface would be complicated, due to both the intrinsic (e.g., grain size, crystalline) and external (e.g., space weathering, imaging conditions, and surface relief) factors. For example, the pervasive space weathering effects on an airless body, such as the Moon, dampens the absorption features. The instrumental noise also impairs the spectral features. These factors affect the determination of band parameters and mineral abundances. Therefore, the spectral data covering less mature surfaces (e.g., fresh craters) are preferred for extracting the mineralogical information. For spectral unmixing, a priori knowledge of endmembers is sometimes challenging to obtain, or unavailable in practice. The unmixing performance would be impaired when using imprecise endmembers. Extracting in-scene

endmembers will alleviate this issue to some extent. Using an imaging spectrometer to acquire meteorite HSI and test the endmember extraction algorithms will be the focus in our feature research.

## 5. Conclusions

We have investigated the lunar meteorite Northwest Africa 8687 by integrated petrographic and spectral approaches. Our conclusions are the following:

1. NWA 8687 is an anorthositic norite mainly composed of plagioclase (67 vol.%) and mafic silicates (21 vol.% for pyroxene and 12 vol.% for olivine), with minor spinel and accessory minerals, such as Fe–Ni metals, troilites, ilmenites, and baddeleyites.
2. The mineral composition is relatively uniform in NWA 8687, with plagioclase dominated by plagioclase (Ca-rich feldspar; $An_{95.6-97.5}$), and olivine dominated by forsterite (Mg-rich olivine; $Fo_{75.8-77.7}$). The main component of pyroxene is enstatite (low-Ca pyroxene; $En_{65-78}$ $Fs_{17-20}$ $Wo_{3-20}$), followed by augite (high-Ca pyroxene; $En_{43-50}$ $Fs_{10-16}$ $Wo_{38-43}$). A Mg-spinel (pink spinel) occurs in the sample, with Mg# > 65 and Cr# < 10.
3. Mineral compositions and bulk chemistry imply that NWA 8687 has characteristics of lunar Mg-suite rock, and is likely paired with the NWA 5744 clan, although their petrological classification is slightly different.
4. The mafic mineral compositions inferred from VNIR reflectance are consistent with the petrological result. However, it is challenging to determine plagioclase, since mafic minerals easily mask its spectral features.
5. The mineral modal abundance of the NWA 8687 chip derived from linear unmixing confirms the noritic characteristics of the sample. It shows that spectral investigation has the potential to classify lunar highland rocks, although extended works of diverse lithologies are warranted.

**Supplementary Materials:** The following supporting information can be downloaded at: https://www.mdpi.com/article/10.3390/rs14122952/s1, Tables S1–S5: Major element abundances of minerals in NWA 8687.

**Author Contributions:** Conceptualization, L.Q. and X.W.; methodology, L.Q., X.W. and L.H.; software, L.Q., X.W. and L.H.; validation, L.Q. and X.W.; formal analysis, L.Q. and X.W.; investigation, L.Q. and X.W.; resources, Y.Z.; data curation, L.Q., X.W. and L.H.; writing—original draft preparation, L.Q. and X.W.; writing—review and editing, L.Q., X.W. and Y.L.; visualization, L.Q., X.W. and L.H.; supervision, X.W., Y.L. and Y.Z.; project administration, Y.L. and Y.Z.; funding acquisition, Y.L. and Y.Z. All authors have read and agreed to the published version of the manuscript.

**Funding:** This research was funded by National Key R&D Program of China (No. 2020YFE0202100) and the pre-research project on Civil Aerospace Technologies funded by China National Space Administration (CNSA) (Grants No. D020201 and D020203). Xing Wu also acknowledges the support from the China Postdoctoral Science Foundation (No. 2021M700149).

**Data Availability Statement:** The RELAB spectral database is available on PDS Geosciences Node Spectral Library (https://pds-speclib.rsl.wustl.edu/ (accessed on 1 May 2022)).

**Acknowledgments:** We acknowledge Yuchen Xu for providing the sample, and we are grateful to Yazhou Yang for his valuable discussions. Many thanks to Xiaohong Mao (Chinese Academy of Geological Sciences) for her kind assistance with the EPMA.

**Conflicts of Interest:** The authors declare no conflict of interest.

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
