# Peer review of "Spectroscopic and Petrographic Investigations of Lunar Mg-Suite Meteorite Northwest Africa 8687"

_remotesensing, doi:10.3390/rs14122952_

Round 1
Reviewer 1 Report
This manuscript gives a detailed petrographic and spectroscopic investigation of a lunar meteorite. The results are compared with several other potentially similar samples and the effectiveness of spectroscopy is also discussed. The manuscript is clearly written and informative. It is essential to constantly enrich laboratory measurements to better constrain remote sensing interpretations.
I would recommend that the authors explicitly include a discussion about the connections between lab results and remote sensing interpretations, which is certainly appropriate since the manuscript is submitted to a remote sensing focused journal. The authors can highlight the implications of this study on remote sensing geologic mapping and potential uncertainties.
Other questions/comments:
Is there any previous analysis on NWA 8687? Are there any differences between that and the authors’ results?
Fig. 10: I can see a considerable shift between the modelled band center near 1900nm and that of the measured spectra for all samples.
Fig. 13: Would the deviation in classification (petrography vs spectroscopy) lead to considerable differences in e.g., regional geologic interpretations?
Reviewer 2 Report
The authors provided a manuscript on “Coordinated Spectroscopic and Petrographic Investigation of Lunar Mg-suite Meteorite Northwest Africa 8687”.
However, this manuscript has several major issues which should be addressed before consideration for publication.
In their introduction, the authors argue that the rarity of extraterrestrial samples, non-destructive methods need to be explored for future analysis. To this end, spectral investigations of the lunar meteorite North West Africa (NWA) 8687 were performed and petrographic study was performed to better characterize the sample. However, most of the research (significant analyzes and new findings) is focused on petrology. I think this paper should be submitted to a petrology journal rather than Remote Sensing journal.
The remote sensing unmixing technique was not used properly. It can be seen that the spectrum of the NWA 8687 sample is very similar to that of OPX, even without the use of unmixing technique. Moreover, as the authors noted, it does not accurately estimate the abundance due to the masking effect. Do you need an unmixing technique to simply distinguish between noritic and troctolitic? It is hard to believe that spectroscopy has contributed significantly to the new findings or conclusions of this study. Also, how this method could be extended to remote sensing research was not specifically discussed.
Please refer to the attached file including specific comments and suggestions.

Reviewer 3 Report
The manuscript reviewed (id remotesensing-173189) is entitled "Coordinated Spectroscopic and Pertrogaphic Investigation of Lunar Mg-suite Meteorite Northwest Africa 8687".
The authors studied the lunar meteorite Northwest Africa (NWA) 8687 and performed a spectroscopic and petrographic investigation on a chip in order to characterize the composition of the sample using a non-destructive approach.
The scientific work and results presented in this manuscript are interesting mostly due to the combined spectroscopic and petrological approach to study a meteorite sample. Although, the processing techniques that were used (e.g. SAM, spectral unmixing) are not innovative, it is interesting to see their application and corresponding results on a lunar meteoritic sample.
Specific comments/suggestions:
Line 34: "global remotely sensed data reveals", correct to "reveal" (data is the plural form of datum)
Paragraph 2.2: Please provide in the text the number of measured spectral bands.
Methodology:
Paragraph 2.4.1.: Why did the authors apply continuum-removal using the entire spectrum and not focus on applying separate continuum-removal around 1μm and 2μm features in order to obtain more accurate band centers, band positions and Band Area Ratio to these features?
Before paragraph 2.4.2, they authors should add a paragraph with the SAM approach which was used for qualitative analysis and was presented for the first time in the Results section Lines 266-288.
Lines 169-170: What do they authors mean by “carefully selected” endmembers? Did they use the RELAB ones with the lowest SAM result? Please elaborate on the selection of endmembers for spectral unmixing in paragraph 2.4.2. In the case of RELAB spectra, please also reply to the question for Lines 275-288.
Line 2.4.2. Fully constrained unmixing is usually used when there is no doubt that the only spectral signatures (materials) existing in the unmixed spectrum are those inserted as endmembers. Is this indeed the case here? Are the authors sure of that? Please discuss.
Lines 275-288: How did the authors compare (using SAM) their spectra with those of RELAB which are measured by different spectrometers and probably have different spectral band centers and number of spectral bands? Did they use response functions to convert one type to another? Or did they follow another approach? I suggest to add a relative text in the manuscript.
Lines 278-279: What threshold angle did the authors define in SAM? What were the angles provided by SAM for each spectrum that provided the best results?
Concluding, providing that the authors take into consideration the aforementioned suggestions and perform the corresponding corrections, I would be in favor for the publication of this manuscript.
Reviewer 4 Report
The work presents the results of studying lunar meteorite composition with petrographic and spectral methods. In general, it was shown that results obtained using VNIR technique corresponded to those obtained by petrographic study. Thus, VNIR technique may be potentially used for further studies of lunar rocks, which in turn have big role in the understanding of Moon geology evolution.
As the results were obtained using appropriate techniques, while the conclusions have been supported by the results, the work may be accepted without significant revisions.
Author Response
We gratefully thank the reviewer for the positive evaluation and the recommendation of publishing this manuscript in Remote Sensing.